# Induction of Somatic Embryogenesis in Plants: Different Players and Focus on WUSCHEL and WUS-RELATED HOMEOBOX (WOX) Transcription Factors

**DOI:** 10.3390/ijms232415950

**Published:** 2022-12-15

**Authors:** Marco Fambrini, Gabriele Usai, Claudio Pugliesi

**Affiliations:** Department of Agriculture Food and Environment, University of Pisa, Via del Borghetto 80, 56124 Pisa, Italy

**Keywords:** somatic embryogenesis, totipotency, *WUSCHEL-related homeobox* genes, transcription factors, epigenetic changes, plant growth regulators, reprogramming cell fate

## Abstract

In plants, other cells can express totipotency in addition to the zygote, thus resulting in embryo differentiation; this appears evident in apomictic and epiphyllous plants. According to Haberlandt’s theory, all plant cells can regenerate a complete plant if the nucleus and the membrane system are intact. In fact, under in vitro conditions, ectopic embryos and adventitious shoots can develop from many organs of the mature plant body. We are beginning to understand how determination processes are regulated and how cell specialization occurs. However, we still need to unravel the mechanisms whereby a cell interprets its position, decides its fate, and communicates it to others. The induction of somatic embryogenesis might be based on a plant growth regulator signal (auxin) to determine an appropriate cellular environment and other factors, including stress and ectopic expression of embryo or meristem identity transcription factors (TFs). Still, we are far from having a complete view of the regulatory genes, their target genes, and their action hierarchy. As in animals, epigenetic reprogramming also plays an essential role in re-establishing the competence of differentiated cells to undergo somatic embryogenesis. Herein, we describe the functions of WUSCHEL-RELATED HOMEOBOX (WOX) transcription factors in regulating the differentiation–dedifferentiation cell process and in the developmental phase of in vitro regenerated adventitious structures.

## 1. Introduction

Higher plants retain the capacity of unlimited growth that relies on the mitotic activity of cells located in the apical and root meristems, which continuously divide to renew themselves and produce cells for organ formation [1]. Plant developmental processes are very flexible compared to those described in animals. Hydra, planarians, and echinoderms preserve interstitial cells and neoblasts (Figure 1), which can replace differentiated cells lost after injury [2].

The observation that genes critical for proper embryonic development in higher animals, such as those coding for T-domain transcription factors or involved in Wnt/Wingless signaling, are expressed during de novo *Hydra* head regeneration has led to important insights into the molecular basis of animal self-organization [3]. Furthermore, some differentiated cells can switch their fate to acquire a new one (trans-differentiation): in axolotls (Figure 1), neural cells can trans-differentiate into muscle and cartilage [4], and zebrafish can regenerate their hearts fully, even after 20% of the ventricle is removed [5,6]. In mammals, trans-differentiation can involve liver cells [7]. Nevertheless, in the animal kingdom, only the zygote can produce a mature organism that requires the ability to generate all the cells of the body, as well as to organize them into a specific temporal and spatial sequence, that is, to undergo a coordinated process of development. In vertebrates, virgin birth is the development of an embryo from a female gamete (parthenogenesis), an unusual process [8,9,10,11]; by contrast, clonal reproduction is more frequently observed in several invertebrates [12,13,14]. Therefore, totipotency, in this strict sense, is established by the ability of an isolated cell to produce a fertile, adult individual [15].

In plants, the zygote is not the only cell that can develop into an embryo. Zygotic embryogenesis depends on fertilization, but apomictic embryos produced in certain species provide clear evidence of in vivo embryo development without sex [16,17,18,19,20,21]. In epiphyllous species (Figure 1), even somatic cells of vegetative organs can spontaneously form in vivo ectopic embryos [19,22,23,24,25,26,27,28]. For a long time, the first hints about how plant cells might regulate totipotency came from empirical approaches based on in vitro culture methods. In 1902, Haberlandt proposed the concept of plant cell totipotency [29], hypothesizing that entire plants could be generated from somatic cells. Experimental evidence supporting this hypothesis was lacking until 1958, when Steward et al. [30] demonstrated that segments of differentiated secondary phloem tissue from carrot could regenerate whole plants, thus highlighting the remarkable totipotent potential of somatic cells. This result was shortly confirmed [31].

The differentiation process leads to a range of determinate cellular fates, which are functional to several organs of the mature organism. The functions are performed efficiently and to the benefit of the whole organism, but at the price that the specialized cells have only limited parts of their genome open for transcription. Nevertheless, cell commitment is reversible, and new developmental patterns can arise as a response to external or internal factors [32,33,34,35,36,37,38,39,40,41]. The recurrent ontogenesis presumes that each differentiated cell can restore the morphogenetic potential of the zygote. This feature is evident during in vitro regeneration when the positional signals are strongly altered, and the differentiated cells can suddenly change identities and follow a gradual process of de-differentiation [38,42]. Thus, differentiated cells can completely revert to the meristematic state or, in some cases, to the somatic embryogenic one (totipotency) [39]. The mechanisms allowing the conservation of the stem cell niche and the change of cellular fate are only partially understood. Nevertheless, it has been demonstrated that somatic embryogenesis can be achieved by ectopic overexpression of some genes, which is crucial for the formation and maintenance of the shoot apical meristem (SAM) and embryo development [43,44,45]. Among them are those encoding the homeodomain transcription factors (TFs) WUSCHEL (WUS), AINTEGUMENTA-LIKE (AIL) AP2/ERF-domain TF PLETHORA4/BABY BOOM (PLT4/BBM) or PLT5/EMBRYO MAKER (PLT5/EMK), MADS-box TF AGAMOUS-LIKE15 (AGL15), NF-Y (nuclear factor of the Y box) TF LEAFY COTYLEDON1 (LEC1), B3 TF LEC2 and FUSCA3 (FUS3), MYB TF MYB118 and RWP-RK DOMAIN-CONTAINING4 (RKD4)/GROUNDED (GRD), class I KNOX homeodomain SHOOT MERISTEMLESS1 (STM1), and the receptor kinase CLAVATA1 (CLV1) [46,47,48,49,50,51,52,53,54,55,56,57,58,59,60,61,62,63,64,65]. In addition, a large number of other TFs that are differentially activated during somatic embryogenesis have been identified through time-course analyses (e.g., Somatic Embryogenesis Receptor Kinase (SERK), CUP-SHAPED COTYLEDON1 (CUC1) and CUC2, STM-Like [66,67,68,69]. The broad range of regulatory factors involved in acquiring embryogenic competence suggests that numerous independent and interrelated pathways are functional in this process. Whether and how these genes work together to promote embryogenic cell formation is unknown [70]. It is likely that epigenetic changes can decide the cell fate, and that they constitute a layer of control superimposed on the activity of the plethora of TFs which are involved in these processes [71]. For example, several reports have suggested that epigenetic mechanisms may have an essential role in cellular de-differentiation, changing cell fate, and in plant totipotency [36,39,45,72,73,74].

Although much is known about the patterning of the shoot meristem during embryogenesis, there is little understanding of patterning that must occur during de novo induction of plant tissues in culture. For example, by overexpressing *WUS*, the formation of somatic embryos (SEs) from zygotic embryos and vegetative tissue was enhanced. These SEs can develop normally and germinate into typical plants [70]. However, how *WUS* regulates the identity of totipotent cells directed toward SE initiation and development needs to be better understood. In this review, we discuss the role of WUS-related homeobox (WOX) TFs in the induction of cell totipotency as well as the regulation of somatic embryogenesis, also considering the interactions of WOX with other TFs, plant growth regulators (PGRs), and epigenetic signals.

## 2. WUS-Related Homeobox (WOX) Genes

### 2.1. Nomenclature and Molecular Characteristic Motifs

Genes codifying for homeobox proteins contain a widely conserved domain of 180 bp, which encodes for a globular homeodomain (HD) of approximately 60 amino acids usually involved in DNA binding [75,76]. Likely, the first *homeobox* genes could have appeared during early eukaryote evolution, probably deriving from a helix-turn-helix (HLH) TF [77]. In plants, the first *homeobox* gene was identified in the early 1990s. Vollbrecht et al. [78] resumed the study of an old maize mutant named *Knotted-1* (*Kn-1*) in which clusters of cells located along the lateral leaf veins continued to divide, forming characteristic growths known as knots [79].

The *WUS-related homeobox* (*WOX*) genes encode for a class of plant-specific homeobox TFs, which perform several key functions in plant development processes. These include, for example, organization and embryonic development, maintenance of stem cells, and formation of various organs [1,80,81,82,83,84,85,86]. These functions may be related to the increase in cell division and/or to the prevention of early cell differentiation. In *Arabidopsis*, 15 different *WOX* genes were identified. An analysis obtained by a phylogenetic tree divided the WOX TFs into three main clades: WUS, INTERMEDIATE, and ANCIENT [87]. The specific function of each *WOX* gene is determined by spatiotemporal expression patterns and by the interaction of their products with other proteins [88]. For example, *WUS* acts mainly as a repressor in stem cell populations in shoot meristems (SAM) formation, but becomes an activator when involved in the regulation of the *AGAMOUS* (*AG*) gene in flower development [80,87,89,90,91,92].

In WOX sequences, the helix-loop-helix-turn-helix (HLHTH) motif is characterized by three α-helices joined by a short loop and a short turn (Figure 2A,B).

The second helix binds DNA through hydrogen bonds and hydrophobic interactions. In particular, specific regions of the protein chains establish a bond with the exposed bases and with thymine methyl groups within the major DNA groove. Analysis of the tertiary structure of sunflower (*Helianthus annuus*) HaWUS proteins revealed highly conserved structures compared to the homeodomain of other species (Figure 2C) [84,88,93].

In the carboxy-terminal region, members of the WUS clade also contain the WUS-box motif T-LX-LFPXX (where T stands for Threonine, L for Leucine, F for Phenylalanine, P for Proline, and X for any amino acid), which distinguishes them from other homeobox TFs (Figure 2A). The WUS-box motif is essential for WUS functions, regulating the stem cell population of SAM and controlling flower development [88,89]. In addition, members of the WUS clade possess a motif of two amino acids (Threonine and Leucine) at the beginning of the WUS-box, whereas the WOX proteins belonging to the other clades show amino acid variations in the beginning position. Some WOX proteins contain a series of amino acids between the HD and the WUS-box that could potentially act as an activation domain, and/or show a carboxy-terminal ERF-associated Amphiphilic Repression (EAR) domain (Figure 2A) [88,93,94,95,96,97,98,99,100,101,102,103]. The EAR motif structure identified in *Arabidopsis* is [LVI]-X-[LVI]-X-[LVI] (where V represents Valine). This motif has been detected for all three major WOX clades members [87]. The C-terminal EAR domains in WUS, WOX5, and WOX7 regulate the repression activity in vegetative and floral meristems [89]. This can be partially mediated by interaction with other proteins, such as TOPLESS, which interacts with WUS through the EAR motif [89,104,105]. In *Oryza sativa*, WOX11 and WOX3 do not possess the EAR motif; nevertheless, they can act as repressors. For instance, WOX11 directly represses the transcription of the *Two-component response regulator OsRR2* (*RR2*) gene, which encodes for a TF that acts as a negative regulator of cytokinin (CK) signaling [106]. In rice, WOX3 can block the gene involved in synthesizing the YABBY3 (YAB3) TF during leaf development, controlling the abaxial cell fate [89,107]. In addition, a simple L-X-L motif has been detected in all *Arabidopsis* WOX proteins, except for WOX8 and WOX10 [87]. Most members of the WOX family work as TFs; however, no Nuclear Localization Signals (NLS) have been predicted for any of them. Nevertheless, WUS, WOX6 [108], and WOX11 [106] are located inside the nucleus. This nuclear localization could indicate NLS motifs which have not yet been detected, or the interaction with other proteins that support NLS function.

### 2.2. In Vivo Roles of WOX Genes in Plants: Developmental Aspects and Stress Response

In *Arabidopsis*, *WUS* and *WOX5* genes maintain stem cell functioning in SAM cells and root apical meristem (RAM) [109]. In particular, expression of *WUS* begins in the 16-cell stage embryo in two inner apical cells, and maintains a tightly restricted pattern throughout embryogenesis [80,110,111]. *WOX5* is also expressed in the early stages of lateral root and cotyledon development [112]. In the case of *WUS* inactivation, totipotent cells, whose fate is regulated by signals deriving from the Organization Center (OC) of the SAM, underwent differentiation in both *Arabidopsis* and *Antirrhinum majus* [110,113]. In *Arabidopsis*, maize, and rice, *WUS* also controls the development of ovules and anthers [114,115]. *wox5* mutants of *Arabidopsis* showed that totipotent cells of columella root undergo differentiation processes [109]. *WOX3* of *Arabidopsis* and the orthologous *NARROW SHEATH 1* (*NS1*) and *NS2* of *Zea mays* regulate the recruitment of initial organ cells in response to signals from peripheral regions of meristems, promoting cell proliferation [116]. In *Arabidopsis*, *WOX6* prevents premature differentiation during the formation of the integuments that envelop the embryonic sac and the egg cell [108]. A further role of *WOX6* has been identified through the isolation of the *hos9-1* mutant, characterized by slower growth and late flowering, with an accentuated sensitivity to low temperatures that leads the plant to undergo extensive freezing phenomena [117]. In *Arabidopsis*, *WOX2* is regulated by *WOX8* and *WOX9,* and is required to form the SAM during embryonic development [118]. *WOX1*, *WOX3*, *WOX5, WOX8,* and *WOX9* are redundantly expressed together with *WOX2* during SAM formation [118]. In the early stages of plant development, *WOX2* and *WOX8* are co-expressed in the zygote. However, during embryonic development, the expression of *WOX2* is restricted in the apical zone, while *WOX8* and *WOX9* are restricted to the basal region [119]. In fact, the products of the *WOX8* and *WOX9* genes are required in large quantities in *Arabidopsis* to develop the micropylar region of the embryo in suspensor and hypophysis organization. Therefore, the molecular system controlled by *WOX* genes establishes the central axis of the embryo and regulates the localized response to auxin by interacting with the auxin transporter PIN-FORMED1 (PIN1) [88]. *WOX9* is also strategic for SAM maintenance [120] and cell division activity during embryonic and post-embryonic development in *Arabidopsis* [121], *Solanum esculentum* [122], and *Petunia hybrida* [123]. In *Arabidopsis*, the genes of the ANCIENT clade *WOX13* and *WOX14* are expressed in primary roots, lateral roots, and floral organs, precluding premature cell differentiation [124]. The development of an adaptable root system is essential for firm crop production under variable environments. In rice plants, two lateral roots have been described: S-type (short and thin) and L-type (long, thick, and capable of further branching). Recently, it was evidenced that two *WOX* genes have opposing roles in controlling LR primordium (LRP) size in this crop [125].

In a postulated model that regards the SAM, vegetative stem cell homeostasis is controlled by the interaction between *WUS* and the *CLV* genes, which are required to rapidly downregulate *WUS* in apical daughter cells after cell division [126,127,128,129,130]. WUS promotes the expression of *CLV3*, which, via activation of the signaling pathway dependent by the CLV1/CLV2 protein complex, acts on *WUS* transcription and limits the size of the WUS-expressing OC [126,131]. The ability of the protein WUS to migrate to adjacent cells and activate its negative regulator is unique to plant stem cell niches [83]. An additional negative feedback mechanism, implying both *WUS* and *AG* genes, is responsible for the maintenance of floral stem cells. In particular, WUS activates the transcription of the *AG* gene, which is responsible for the identity of the flower organs. Later, *AG* represses the *WUS* transcription with negative feedback [132]. In *Arabidopsis* roots, the signal peptide CLAVATA3/EMBRYO SURROUNDING REGION40 (CLE40) and ARABIDOPSIS CRINKLY4 kinase (ACR4) (a member of the receptor-like kinase family CRINKLY4) have been identified as regulators of *WOX5* activity, with a negative loop mechanism similar to that involving WUS and CLV [133]. It has also been shown that WUS can repress the transcription of some *Type-A Arabidopsis Response Regulators* (*ARR-A*) genes, which encode negative regulators of signaling mechanisms originating from the CK [134]. ARR-A proteins probably compete for phosphorylation with positive *ARR-B* regulators [81].

Recently, the role of *WOX* genes has been extended in bulbil differentiation, a vegetative propagation strategy of *Lilium lancifolium* [135]. *LlWOX9* and *LlWOX11* are positive regulators of bulbil formation. Cytokinin type-B LlRRs can bind to the promoters of *LlWOX9* and *LlWOX11* to promote their transcription. In addition, *LlWOX11* can enhance cytokinin signaling by inhibiting the transcription of type-A LlRR9 [135]. In addition, the asexual propagation in three Crassulaceae species showed the concomitant expression of *WUS* with the shoot induction phase on leaves [136].

In plants, tissue repair is essential to counteract damage-associated stress; moreover, repair processes are required in grafting events. Recently, the involvement of WOX13, an ancient member of the WOX family, in callus formation and organ adhesion in *A. thaliana* has been reported [137]. After wounding, WOX13 expression was rapidly induced, and this response was partly dependent on the activity of WOUND-INDUCED DEDIFFERENTIATION 1 (WIND1). Interestingly, WOX13 directly upregulated WIND2 and WIND3 to reinforce cellular reprogramming and organ regeneration [137]. To elucidate the role in callus formation and tissue repair of *WOX13*, a schematic diagram was reported by Ikeuchi et al. [137].

Recently, constructive research has finally interpreted the molecular mechanism of meristematic cell resistance to viruses, an essential phenomenon in the health restoration of crops [138,139]. WUS responds to cucumber mosaic virus (CMV) infection and blocks virus accumulation in the central and peripheral zones of the meristem. In particular, WUS hinders viral protein synthesis by repressing the expression of plant *S*-adenosyl-L-methionine–dependent methyltransferases with a role in ribosomal RNA processing and ribosome stability [138].

Environmental factors can significantly influence the development and final yield of many crops. To cope with environmental stresses, plants have evolved various defense mechanisms. At the gene level, one of the most effective mechanisms is the regulation of specific genes encoding TFs [140]. Among these, it has been shown that *WOX* genes also play a key role in response to multiple types of stress. For example, in rice, overexpression of the *OsWOX13* gene under the control of the *rab21* promoter increases the tolerance to drought [141]. In *Jatropha curcas*, water deprivation strongly reduces the expression of both *JcWOX5* and *JcWOX6* genes, while intense saline stress induces the expression of *JcWOX1* and *JcWOX8* [140]. In *Gossypium hirsutum*, over half of *WOX* genes show no significant response to heat, cold, salinity, or drought. By contrast, the *GhWOX10_Dt*, *GhWOX13a_At/Dt,* and *GhWOX13b_At/Dt* genes are strongly induced under multiple stress [94]. In the hybrid *Populus alba* × *P. glandulosa*, the *PagWOX11/12a* gene is predominantly expressed in roots under drought [142]. In *Camellia sinensis*, the lack of water activates *CsWOX13* and *CsWOX15,* but inhibits the expression of *CsWOX1* and *CsWOX9*. Tea plants subjected to cold conditions showed reduced expression of both *CsWOX9* and *CsWOX14*. Furthermore, the *CsWOX5*, *CsWOX3,* and *CsWOX2* genes are significantly regulated by exogenous treatments with ethylene, abscisic acid, methyl-jasmonate, and gibberellic acid, respectively [143]. In *Brassica napus*, 58 *WOX* genes have been identified and characterized. Analysis of their expression levels showed that four members belonging to the *WOX4* sub-clade (*BnWOX10*, *BnWOX50*, *BnWOX44,* and *BnWOX18*) are activated by abiotic stress or PGR treatments [95].

## 3. *WOX* Genes Activity during Embryogenesis and Interaction with Plant Growth Regulators

### 3.1. Zygotic Embryogenesis

Different embryogenic processes share a common PGR control that essentially relies on the action of auxin, abscisic acid (ABA), cytokinins (CKs), and gibberellins (GAs). In particular, the establishment of auxin synthesis and polar auxin transport (PAT) are key steps in embryo development [144]. During early zygotic embryogenesis of *Arabidopsis*, auxin accumulates dynamically at specific positions that correlate with developmental cell fate. In such a way, the developmental decision is tied to both auxin transport regulators and components of response machinery [145]. Members of the *PIN FORMED* (*PIN*) family of auxin efflux facilitators are the key players in determining the direction of intercellular auxin flow [146,147,148]. After zygotic division, PIN7 localizes at the upper side of basal cell descendants and reverses orientation at the globular stage. The localization of PIN1 becomes polarized from the midglobular stage onward. These localizations correlate with an auxin maximum in the apical cell at the early stages of embryogenesis, as well as relocation of this maximum to the basal pole at the globular stage. From the embryo’s late-globular stage onward, PIN4 is detected along the surface of the hypophysis and at the lower side of the adjacent suspensor cell [146]. The polar localization of PIN1 is disrupted in the *gnom* mutant embryos that do not undergo coordinated development [149]. *GNOM* is required for PIN1 recycling between the plasma membrane and endosomes [145]. In extreme cases, loss of multiple PIN family members phenocopies *gnom* embryos. When the endocytosis-dependent mechanism of PIN polarity generation is altered, auxin response in apical embryo regions increases, leading to a cell fate change from cotyledon to root [147]. In *A. thaliana*, specific dynamics of *WOX* gene expression identified cell fate decisions during zygotic embryogenesis. For example, *WOX2* and *WOX8*, initially coexpressed in the zygote, act as complementary cell fate regulators in the apical and basal lineage, respectively, during the apical–basal axis formation. *WOX2* expression in the apical lineage is an early and important downstream function of *WOX8*/*WOX9* activity, and the *WOX* transcriptional machinery is linked with the establishment of a localized auxin response in the proembryo [118]. Moreover, the expression of *WOX8* is independent of the axis patterning signal auxin; by contrast, with the redundant gene *WOX9*, it is activated in the zygote, its basal daughter cell, and the hypophysis by the zinc-finger transcription factor WRKY2 [150]. It is interesting to note that an ancestral role of WOX in seed plant embryo development has been demonstrated, thus corroborating the proposed connection between PAT, PIN-FORMED (PIN), and WOX in regulating embryo patterning in seed plants [151].

Analysis of the expression pattern of *WOX* genes in maize during zygotic embryogenesis suggested that apical and basal cell lineages and cell fate determination in grasses may involve different players, or occur at a later embryonic stage compared to *A. thaliana* [152].

In *Picea abies* zygotic embryogenesis, Zhu et al. [153] revealed that *PaWOX2* plays a more crucial role during early embryogenesis (i.e., protoderm formation and suspensor expansion) than during late embryogenesis. This essential function was evident because the downregulation of PaWOX2 at the beginning of embryo development caused a significant decrease in the yield of mature embryos. By contrast, the downregulation of PaWOX2 after late embryos were formed did not affect further embryo development or maturation [153].

### 3.2. Somatic Embryogenesis

Somatic cell reprogramming and transition to embryogenic cell competence also require high concentrations of auxin (primarily 2,4-dichlorophenoxyacetic acid (2,4-D), which is a synthetic auxin-like plant growth regulator), which presumably acts to trigger a signaling cascade in the modulation of numerous SE-associated TF genes [34,39,144]. A transient increase in endogenous auxin (IAA) levels has been observed in various systems [154]. For example, 2,4-D must be removed from the culture medium to induce SE formation from the embryonic callus. The removal of 2,4-D activates the expression of *YUCCA2* (*YUC2*) and *YUC4*, which encode auxin biosynthetic enzymes, resulting in increased endogenous IAA levels [155,156]. In turn, the different expression of the genes encoding the core components of the auxin-signaling pathway, the AUXIN/INDOLE-3-ACETIC ACIDs (Aux/IAAs), and AUXIN RESPONSE FACTORs (ARFs), was demonstrated to accompany SE induction. In particular, a crucial role of asymmetric auxin distribution has been established for de novo morphogenesis in in vitro SEs using live imaging analysis of *PIN* gene expression [157].

In *Arabidopsis*, the results of Su et al. [155] established that an auxin gradient and PIN1-mediated PAT are essential for *WUS* induction and somatic embryogenesis. Su et al. [155] also showed that other regulatory genes of zygotic embryogenesis were upregulated during SE development. Nonetheless, *WUS* expression was identified within the embryonic callus when SEs could not be morphologically recognized. In addition, Su et al. [158] demonstrated that the expression patterns of several regulatory genes critical for RAM formation were correlated with the establishment of the embryonic root meristem during somatic embryogenesis in *Arabidopsis*. Notably, early expression of *WOX5* and *WUS* genes was induced, and nearly overlapped within the embryonic callus when SEs could not be identified morphologically. In addition, cytokinin response signals were detected in specific regions correlated with induced *WOX5* expression and subsequent SE formation. Overexpression of *ARR7* and *ARR15* (feedback repressors of cytokinin signaling) disturbed RAM initiation and SE induction. These results provided information regarding auxin and cytokinin-regulated apical-basal polarity formation of the shoot–root axis during somatic embryogenesis [158].

In *Medicago truncatula,* a comparative analysis of the expression of *WOX* and *PIN* genes in ovules, and of the course of somatic embryogenesis, was performed [159]. *MtWOX11*-*like* and *MtPIN10* showed an increased expression level in ovules, and were activated during SE development. By contrast, in other *WOX* and *PIN* genes, the expression level was low in ovules and did not show transcription activation associated with somatic embryogenesis. These results confirmed that the exact regulatory mechanisms could control the early stages of somatic and zygotic embryogenesis [159].

In *Picea abies*, *PaWOX2* expression was highest during embryogenesis at the earliest stages of development. However, no activity was detected in non-embryogenic cell culture, indicating that *PaWOX2* plays a fundamental role in early SE development and can be used as a possible marker for embryogenic potential [160]. It has also been shown that the PAT inhibitor NPA impairs embryo morphology and increases the expression of *PIN1* during *Picea abies* SE development [161]. The results, in Gymnosperms as well, strengthen the proposed connection between PAT and WOX in the regulation of embryo patterning in seed plants [161,162]. The close link between PGRs and *WOX* genes is expected to induce somatic embryogenesis. In *Triticum aestivum*, *TaWOX5* was primarily expressed in the root and calli induced by auxin and cytokinin, indicating that *TaWOX5* may be related to root formation or development, and is associated with regulation of PGRs in somatic embryogenesis [163]. In the interspecific hybrid *Liquidambar styraciflua* × *Liquidambar formosana*, auxin, cytokinin signal transduction, and biosynthesis-related genes were significantly expressed during somatic embryogenesis. Among these, there were many auxin signal transduction genes. In particular, *small auxin up RNA* (*SUAR*) family genes were predominant [164]. In embryogenic callus, the contents of many auxin-related genes, such as *ARF17, ARF18,* and *AUX1*, were significantly higher than in non-embryogenic callus. In particular, Qi et al. [164] found that *PIN1-like, PIN2, AUX1, ARFs, GH3*, and *SAUR* were dramatically upregulated in embryogenic callus and downregulated during SE development, indicating that the auxin-responsive genes played a critical role mainly in the transition from vegetative to embryogenic competent cells. Analogously, *WOX9* was specifically expressed in embryogenic callus and downregulated during SE development [164]. By contrast, *WOX11* was highly expressed throughout SE phases and almost not expressed in non-embryogenic callus or vegetative organs. Hence, these genes might play an essential role in the somatic embryogenesis of hybrid sweetgum, but at a different stage of SE initiation and development.

Time course experiments have often been designed to identify TFs involved in the induction and development of SEs [66,67,68]. In *Lactuca sativa*, the expression *LsWUS1L* was almost absent in the early stages of in vitro culture (1–4 days) in the presence of cytokinin and auxin (with a 4:1 ratio), but it gradually increased during the following days (7–12) of culture [69]. Since the onset of the organization of adventitious structures was observed from seven to twelve days, the time-dependent increase in the transcript level of *LsWUS1L* could be related to the development of buds and/or SEs, as well as the initial organization of SAMs. By contrast, *LsWUS2L* showed a highly significant increase of transcripts up to four days of in vitro culture (Figure 3).

The sudden increase in the transcriptional activity of *LsWUS2L* appears remarkable because, simultaneously, the histological analysis showed that the groups of poorly vacuolated cells, with large nuclei and rich in cytoplasm, began to be present in the explants (Figure 3). These cells subsequently organized themselves into SEs and adventitious buds. Therefore, these results suggest that *LsWUS2L* could be involved in the induction phase of embryogenic competence [69].

### 3.3. WOX Genes Activation and Shoot Regeneration

For *Arabidopsis*, Gordon et al. [165] induced new shoot meristems from cultured root explants. These authors characterized early patterning during the de novo development of the *Arabidopsis* shoot meristems using fluorescent reporters of known genes and proteins required for SAM development and maintenance. They demonstrated that a small number of progenitor cells initiate the development of new shoot meristems through stereotypical stages of reporter expression and activity of *CUC2*, *WUS*, *PIN1*, *STM*, *FILAMENTOUS FLOWER* (*FIL*), *REVOLUTA* (*REV*), *ARABIDOPSIS THALIANA MERISTEM L1 LAYER* (*ATML1*), and *CLV3*. These genes, also interacting with PGRs (cytokinins and auxins), participate in various stages of initiation and coordinated development of the adventitious meristems. In particular, a functional requirement for *WUS* activity during de novo shoot meristem initiation was detected, demonstrating the interaction of *WUS* with cytokinins [165]. This relationship was confirmed by Dai et al. [166], which demonstrated, by chromatin immunoprecipitation (ChIP) and transient activation assays, that in plant cells cultured in vitro, ARR12 binds to the promoter of *WUS*. Therefore, during shoot regeneration, ARR12 acts as a molecular link between cytokinin signaling and the expression of *WUS*.

Regarding in vitro regeneration methods, likely one of the most common is the induction of adventitious buds by an indirect inductive process involving an intermediate callus phase. During the induction of this process, the auxin–cytokinin crosstalk was frequently crucial. Recently, Zhai and Xu [167] reported in *Arabidopsis* that the tissue structure after callus induction was comparable to that of the root primordium or apical meristem. Interestingly, authors identified the middle cell layer characterized by a quiescent center-like transcriptional identity, and at this site, tissues exhibited the ability to regenerate organs. In particular, WOX5 directly interacts with PLETHORA1 and 2 to promote the *TRYPTOPHAN AMINOTRANSFERASE OF ARABIDOPSIS1* expression for endogenous auxin production. WOX5 also interacts with the B-type ARABIDOPSIS RESPONSE REGULATOR12 (ARR12) and represses A-type ARRs to break the negative feedback loop in cytokinin signaling. Therefore, auxin production and the enhancement of cytokinin sensitivity were required for pluripotency acquisition in the middle cell layer of the callus for organ regeneration [167].

To exemplify the studies in which *WOX* genes are involved in in vitro morphogenesis, a selection is shown in Table 1.

## 4. Overexpression of Genes Encoding Transcription Factors to Increase Transformation Rate

Several reports have strongly suggested that the co-expression of morphogenetic genes, as well as *BBM* and *WUS,* stimulates the growth of the embryogenic callus, resulting in improved transformation frequencies with *Agrobacterium tumefaciens* in transformation-recalcitrant monocot species [174,175,176,177,178,179]. Moreover, *WUS* overexpression has been reported to enhance somatic embryogenesis in dicot and gymnosperm species, such as *N. tabacum* [180], *Coffea canephora* [181], *Picea glauca* [182], and *Medicago truncatula* [65]. To examine the effect of the ectopic expression of three *Arabidopsis WOX2*, *WOX8,* and *WOX9* genes on the regenerative competency of tissues and cells cultured in vitro, Kyo et al. [183] developed a transgenic variety of *Nicotiana tabacum* in which these genes were under the transcriptional control of a chemical-inducible expression system. The authors observed remarkable regeneration of plantlets only in explants derived from the hybrids possessing two transgenes, namely *WOX2* combined with *WOX8* or *WOX9*, but found no regeneration in the segments derived from their parental lines, proving that not all genes of the WOX family act in the same way. *Capsicum chinense* varieties are well known for their unique flavors, and many have exceptional heat. The hottest peppers in the world are members of this species [184]. Unfortunately, *C. chinense* is a recalcitrant species for in vitro morphogenesis, and the development of new biotechnological tools (NBTs) has been hindered. Nevertheless, an in vitro transformation method was successfully obtained in this species via *Agrobacterium tumefaciens* co-cultivation with a system that expresses the heterologous gene *WUS* from *Arabidopsis* [185]. Similarly, in *Gossypium hirsutum*, somatic embryogenesis was significantly improved in *AtWUS*-overexpressing calli compared to control explants [186,187].

More recently, Kadri et al. [188] investigated the effect of the heterologous *Arabidopsis WUS* gene overexpression, under the control of the jasmonate-responsive *vsp1* promoter, on the morphogenetic responses of *Medicago truncatula* explants. *WUS* expression in leaf explants increased callogenesis and embryogenesis without growth regulators. Similarly, *WUS* expression enhanced the embryogenic potential of hairy root fragments.

However, especially in maize, overexpression of *BBM* and *WUS* can compromise the quality of regenerated plants, leading to male and/or female sterility. *Agrobacterium*-mediated T-DNA transformation involves transient T-DNA gene expression within 36–48 h of bacterium infection, followed by stable T-DNA integration into the plant genome [189,190]. This has been demonstrated using *CRE*-mediated excision of a genomic locus flanked by homologous loxP sites, without stable integration of the T-DNA harboring the *CRE recombinase* gene [191]. Therefore, to increase the production of fertile T_0_ plants, Wang et al. [192] used an inducible site-specific *recombinase* (*Cre*) to excise morphogenetic genes (i.e., *WUS2* and/or *BBM*) after transformation, but before regeneration. In addition, the use of developmentally regulated promoters, such as *Ole*, *Glb1*, *End2*, and *Ltp2*, to drive *Cre* enabled the excision of morphogenetic genes in early embryo development and produced excised events at a rate of 25–100%.

Several other methods have been developed to exclude the effect of the stable integration of *BBM* and *WUS*. For example, Hoerster et al. [193] determined that the expression of *Zm-WUS2* alone, driven by the maize *Pltp* promoter (*Zm*-*Pltp_pro_*), was sufficient to induce rapid SE formation from the scutella of immature maize zygotic embryos. Notably, the authors demonstrated that co-infecting with two strains of *Agrobacterium*, one with a *WUS2* expression cassette and the other with a combination of selectable and visual marker cassettes, transformed T_0_ plants that contained only a single copy of the selectable marker T-DNA, without the integration of *WUS2*. Furthermore, the process was optimized by varying the ratio of the two *Agrobacterium* strains and by modulating *WUS2* expression to enable high-frequency recovery of selectable marker-containing T_0_ plants, without the *WUS2* gene.

The increase in transformation efficiency in cereal crops is a prominent goal in plant breeding through biotechnological approaches, and recently, Wang et al. [194] have obtained interesting results. These authors demonstrated that overexpression of the wheat gene *TaWOX5* dramatically increases the transformation frequency of wheat, as well as five other cereal species; the significant result was obtained in different genotypes, and had no apparent deleterious effects after regeneration [194].

## 5. Molecular Players in Plant Totipotency

So far, the nature of totipotency remains a mystery in some aspects [73,195,196]. As reported above, some data suggest that in higher plants, in addition to *WUS/WOX* genes, the ectopic upregulation of many other TFs (e.g., LEC1, LEC2, BBM, FUS3, and AGL15) is involved in the induction of embryo formation from somatic cells [47,49,51,52,197]. One common feature of these genes is their role in promoting embryogenic cell competence in the absence of auxin or stress-related treatments, usually used to establish embryogenic programs on in vitro systems [34,73]. However, the variety of the aforementioned regulatory factors also suggests that numerous independent and/or interrelated pathways can lead to the acquisition of embryogenic competence [35,73]. At the same time, *WUS* overexpression has been associated with somatic embryogenesis [70] and the ectopic proliferation of flower meristems [198]. On the other hand, *WUS* overexpression appears to repress the *LEC1* gene [199], whereas the overexpression of *AGL15* upregulates *SERK1*; furthermore, when *LEC2* is ectopically expressed, an increase in *AGL15* can be observed. Whether and how these genes work together to promote embryogenic cell formation is not yet wholly known [200]. Recently, some regulatory networks have been proposed [73,196,201], and some results suggest that the ectopic expression of these TFs is not necessarily linked to the switch of cell fate towards the same embryogenic competence [196]. The expression of *LEC1*, *LEC2*, *WUS,* and *AGL15* genes in the egg cell or zygote is lacking. Currently, the hypothesis that their overexpression acts indirectly, causing some stress or responses that lead to change cell fate, is still open. In addition, the cellular context in which a gene is expressed appears to be important in determining the ability of a cell to become totipotent [196]. For example, *WUS* functions non-cell-autonomously to maintain pluripotent stem cell identity in the meristem niche, whereas it appears to function cell autonomously when ectopically expressed in a range of somatic cells that become totipotent embryogenic cells [70,110].

Notably, in apomictic *Cenchrus ciliaris*, the *BBM-Like* (*BBML*) gene is naturally expressed in the egg cell [202], and expression of *BBML* under its promoter or an egg-cell-expressed promoter can induce parthenogenesis in sexual grasses and tobacco [197,203,204]. In maize and rice, *BBML* genes are expressed in the zygote shortly after fertilization, and in the sperm cell before fertilization [205,206]. In *Arabidopsis*, Chen et al. [207] showed that *BBM* is expressed transiently in the chalazal region of the ovule and seed, in the embryo starting from the zygote stage, and in the first few dividing endosperm cells. These data suggest early *BBM* functions in the embryo and broader functions during seed development. In addition, ectopic *BBM* expression in the egg cell of *Arabidopsis* and in the dicot crops *Brassica napus* and *Solanum lycopersicon* is sufficient to bypass the fertilization requirement for embryo development [207]. Therefore, it was demonstrated that ectopic *BBM* expression in the egg cell is sufficient to induce haploid embryo development. These data suggest that BBM TF is at the apex of a transcriptional network that promotes asexual embryo development [207]. The *BBML* gene in the egg cell is suitable for inducing parthenogenesis in rice [208]. However, to obtain diploid embryos, the misexpression of *BBML* is insufficient. When genome editing to substitute mitosis for meiosis (*MiMe*) [209,210] is combined with the expression of rice *BBM1* in the egg cell, then a clonal progeny can be obtained that retains genome-wide parental heterozygosity [204]. Moreover, Xie et al. [211], via CRISPR/Cas9, deactivated the genes *OsPO11-1*, *OsREC8*, *OsOSD1,* and *OsMATL* to create apomictic rice plants. The resulting quadruple mutant, Apomictic Offspring Production (AOP), showed a transformation from meiosis to mitosis and produced clonal diploid gametes. However, consistently with previously reports, mutation of *OsMATL* gave rise to low fertility and low haploid or apomictic induction rates [212,213].

It should be clear that from the examples reported, members of the WOX TF family play a pivotal role in the acquisition of embryogenic competence, and they control various stages of SE development. However, these data are insufficient to explain the achievement of embryogenic totipotency. It cannot be the activation of a single gene that reprograms the fate of a differentiated cell towards acquiring all the degrees of freedom representative of the zygotic cell. During differentiation, a less specialized cell type gradually transforms into a more specialized one, which is constrained to a stable morphology, structure, and function [214]. Moreover, the differentiation process reduces the cells’ self-renewal ability and embryogenic totipotency [35,73,196].

## 6. An Epigenetic Hierarchical Network Reprograms Differentiated Cells to Pluripotent/Totipotent Cells

In the classical view of development, the differentiated state of a cell was believed to be terminal and irreversible. This concept has been best exemplified by Conrad H. Waddington’s description of the epigenetic landscape [215]. In many cases, the complex PGR and gene regulation, which leads a zygote to differentiate the various cell types of an organism, is only partially understood. Nevertheless, it is much more challenging to understand the reverse path. The differentiation is a process of differential gene activation and de-activation that cannot involve either gene mutations or stress conditions. Thus, it would not be easy to reconcile such a phenomenon which directly affects de-differentiation towards the reacquisition of embryogenic totipotency. Various stressful conditions are known to be inductive to somatic embryogenesis [34,73]. Perhaps we should ask ourselves what differentiates zygotes from all other plant cells. Why is a somatic cell subjected to severe stress conditions, devastating PGR conditions, or the overexpression of many TFs capable of inducing differentiated cells to reacquire potentialities typical of the zygote? The zygote undergoes extensive self-renewal and can differentiate along multiple cell lineages. Differentiation requires long-lasting changes in gene expression, with differential gene inactivation or activation during the process. In plants and animals, epigenetic mechanisms are essential to controlling the heritable cellular memory of gene expression during development [214,216,217,218,219]. Therefore, it is plausible to hypothesize that activating analogous mechanisms is necessary for a differentiated cell to regain embryogenic totipotency [73,196]. Problems arise here. The degree of differentiation of cells grown in vitro can be very significant. We do not know, even superficially, the epigenetic landscapes traveled by the cells in their differentiation [220,221]. Therefore, the stimuli that we supply to cells, or, more commonly, to organs and tissues, to reacquire embryogenic totipotency can only be empirical (e.g., large amounts of auxin, stress conditions, and overexpression of TFs). We should understand how these treatments intersect with epigenetic changes by redirecting differentiated somatic cells to lose their determination state. All of this is to achieve a condition that allows for the development of SE, in some cases, in the absence of exogenous PGR treatments [24,25,26,27,28,70].

It has been highlighted that no spatial regulation is active in the control of the embryogenic cell pool compared to the pool of stem cells in the meristems, where signals from neighboring cells assure stemness [222,223]. According to this hypothesis, embryogenic cells exhibit relevant structural differences concerning meristematic stem cells, dealing with an increased cell wall thickening, a reduced number of plasmodesmata, and a specific nuclear chromatin architecture [222]. Thus, it is likely that the physical or physiological isolation of somatic cells from their immediate surroundings is coupled with the reprogramming of gene expression profiles, leading to a cell fate switch [73,224].

In this context, the de-differentiation event results from the precise choreography of genes whose transcription is temporally and spatially controlled. Several studies have demonstrated the essential role of chromatin structure in controlling gene transcription [214,225]. As a consequence, chromatin remodeling, which can be accomplished by multiple mechanisms such as histone post-transcriptional modifications and changes in DNA methylation [226], is necessary to couple with cellular de-differentiation and switching of cell fate [36,39,73,196,227]. Thus, it is likely that epigenetic mechanisms control cell fate through superimposed activity on the TFs and PGRs, and may have a role in regulating cell totipotency in plants [71]. In carrot plants, the *LEC1* promoter region showed a reduced level of DNA methylation during somatic embryogenesis, followed by an increase during the transition from embryonic to vegetative growth [228]. In addition, if hypermethylation of a region of 5′-*LEC1* promoter was induced by RNA-directed DNA methylation (RdDM), the gene expression was reduced in the embryogenic cells, indicating a negative correlation between DNA methylation and *LEC1* expression [228].

Characterizing the *PICKLE* (*PKL*) gene has strengthened this hypothesis [229,230,231]. The mutant *pkl* of *Arabidopsis* shows swollen primary roots. Its phenotype is characterized by the postembryonic expression of embryo-specific markers (*LEC1*, *LEC2*, and *FUS3*) in the primary root, and by the ectopic proliferation of SEs. *PKL* encodes for an SWI/SNF member, a putative Chromodomain Helicase DNA-binding protein 3 (CHD3). In roots, *PKL*-mediated downregulation of *LEC* genes depends on the activation of Polycomb group (PcG) genes [232,233]. In eukaryotes, the general mechanisms of PcG protein functions are well-preserved, although, in plants, different players appeared during evolution [234]. PcG and Trithorax (TrxG) complexes control the expression of developmental regulator genes through modifications of the chromatin status, e.g., the deposition of the repressive H3K27me3 and the activator H3K4me3, respectively [233,235,236,237,238,239]. PcG proteins exist in multiprotein complexes; polycomb repressive complex 2 (PRC2) is a crucial regulator of epigenetic states which, via its catalytic subunit E(z), shuts off gene expression by trimethylating Lys27 of histone H3, resulting in H3K27me3 [240]. In contrast, PRC1, via its chromodomain-containing subunit polycomb (Pc), binds H3K27me3 generated by PRC2, resulting in a stable silencing chromatin state [241]. In the *pkl* mutant, the reduced PRC1 protein activity can lead to cell de-differentiation and callus-like tissue formation. Mutations in *Arabidopsis* PRC1-like ring-finger protein genes *AtRING1a* and *AtRING1b* lead to the de-repression of embryonic traits during vegetative growth, and induce the ectopic expression of several key regulatory genes involved in embryogenesis and stem cell activity [242]. PRC2 composition is conserved from humans to plants, but the function of PRC2 during the early stages of plant life has not yet been fully defined. PRC2 is involved in repressing embryo maturation programs during the establishment of vegetative development in *Arabidopsis* [243,244,245]. The absence of PRC2 can induce somatic cells to become totipotent, thereby leading to somatic embryogenesis. Thus, in contrast to mammals, where PRC2 proteins are required to maintain pluripotency and prevent cell differentiation, in plants, PRC2 proteins are required to promote cell differentiation by suppressing embryonic development [246,247]. Mutations in genes coding for PRC2 proteins resulted in mutants for fertilization-independent endosperm (*fie*) or seed (*fis*) formation, respectively, suggesting that the embryogenic program is repressed by chromatin-based gene silencing, and becomes released in response to fertilization [248]. Furthermore, as in animals, the plant zygote is transcriptionally relatively quiescent, and maternally stored products can sustain the first cell divisions; by contrast, downregulation of transcription is deleterious to endosperm development. The transient zygotic quiescence, resulting from endosperm-embryo interactions, is probably required for extensive genome reprogramming after fertilization [249].

*WUS* regulation can also be subjected to epigenetic mechanisms [250,251,252]. Based on the observation that the gene *AtGCN5*, coding a histone acetyltransferase, is required to control floral meristem through the *WUS/AG* pathway, it has been supposed that the histone acetylation possibly restricts the *WUS* expression domain within the floral meristem, turning on a *WUS* repressor [253]. Acetylation, a post-translational modification affecting H3 and H4 histones, is performed by highly specific enzymes identified in plants and animals [254,255]. The histone modification system interplays and crosstalks with DNA methylation status, affecting the degree and location of histone post-translational modifications [256,257]. In the promoter region, the methylation of DNA directly lays down the silencing of transcription [258]. In animals, DNA methylation concerns cytosines placed in the dinucleotide CpG, but in plants, the CpHpG and CpHpHpG patterns can also be involved. DNA methylation, catalyzed by DNA methyltransferases (DNMTs), can be reversed by the Ten-Eleven Translocation (TET) proteins in mammalian [259] and the Demeter (Dme) family of DNA glycosylases in plants [260]. In plants, DNA methylation can be directed by small RNAs (RdDM) using two plant-specific RNA polymerases-PolIV (NRPD1) and PolV (NRPE1), and by de novo DNA methylase DRM1/2 “Domain Rearranged Methyltransferase 1/2” [261]. DNA methylations in both CG and CHG sites are achieved by METHYLTRANSFERASE1 (MET1) and CHROMOMETHYLASE3 (CMT3), respectively [262,263,264]. Maintaining methylation in non-CG sites of heterochromatic TEs also requires the activity of CMT2 [265]. Lysine 9 methylation of histone H3 (H3K9me2) is mediated by the SET domain of SU(VAR)3–9 HOMOLOGUE 4/KRYPTONITE (SUVH4/KYP), SUVH5, and SUVH6 proteins, which promotes the binding of CMT3 to chromatin and maintains methylation in CHG sites [263,266]. Furthermore, the activity of DECREASE IN DNA METHYLATION 1 (DDM1), a chromatin-remodeling factor, is required both to modify the chromatin conformation and to maintain the DNA methylation patterns across diverse plant species. This is required for RdDM to maintain mCHH islands [265,267,268].

In sunflower, the involvement of histone H3 methylation in *HaWUS* regulation is also suggested by the presence of several putative binding motives; e.g., PROMO software finds DNA-binding sequences for the ALFIN1-like protein implied in the switch from the H3K4me3-associated active to the H3K27me3-associated repressive transcription state of seed developmental genes, promoting seed germination [84,269]. In *A. thaliana*, a histone acetylase, which is coded by *AtGCN5* and is involved in both long-term and short-term dynamic transcriptional regulations [233,270], limits the domain of *WUS* expression within the floral meristem, turning on a *WUS* repressor through the *WUS/AG* pathway [253]. Moreover, in the termination of floral stem cell maintenance in *A. thaliana*, AG represses *WUS* expression and recruits PcG proteins to deposit H3K27me3 [91] in a still-unknown way.

Based on chromatin accessibility dynamics, Wang et al. [196] showed that the developmental stage of in vitro cultured tissue (i.e., non-germinated seeds) is at the top of the regulatory hierarchy that governs SE initiation. This finding explains why post-embryonic somatic tissues (i.e., germinated seedlings) are resistant to reprogramming for somatic embryogenesis. Wang et al. [196] suggested that the cellular status of the juvenile phase is less amenable to reshaping the chromatin status of the gene loci determining totipotency. It was also hypothesized that histone 2A monoubiquitination at lysine 119 (H2Aub) marking of ABI3/FUS3/LEC2 leads to their initial repression, further maintained by PcG-mediated H3K27me3, and might contribute to the loss of reprogramming competence in the somatic cells after seed germination [196]. In this model, *LEC1*, *BBM,* and auxin form a feed-forward loop to reinforce cell fate transition. *LEC2* acts at the cell totipotent gene network’s output node by activating early embryonic development genes, such as *WOX2* and *WOX3* [196].

In a vision that involves epigenetic mechanisms for controlling cellular totipotency, a key role is also played by micro RNAs (miRNAs or miRs). MiRNAs are the most characterized class of non-coding RNAs, and are engaged in many cellular processes, including cell differentiation, development, and homeostasis [271]. MiRNAs are 21–24-nt-long single-stranded nucleic acids that function post-transcriptionally to regulate gene expression [271,272].

MiRNAs have been implicated in maintaining the pluripotency of mammalian cells [273]. In plants, several types of miRNAs involved in somatic embryogenesis have been reported. For example, *miR156*, *miR162*, *miR166a*, *miR167*, *miR168*, *miR171a/b*, *miR171c*, *miR393*, *miR397,* and *miR398* play a very active role during various stages of somatic embryogenesis [274,275,276,277]. Genome-wide analysis of the somatic embryo transcriptome in *Arabidopsis* indicated that numerous miRNAs are differentially expressed during somatic embryogenesis by the extensive modulation of transcription factor gene expression [66,272,278]. In particular, there are some indications of their role in maintaining pluripotency [73,279]. Among them, *miR160* interacts with *ARF10* to inhibit callus initiation and shoot regeneration. At the same time, *miR156*, which targets *SQUAMOSA* promoter-binding protein-like (SPL) TFs, contributes to a decline in the explants’ capacity to regenerate shoots [280,281,282]. *MiR160*, targeting *ARF10*, *ARF16*, and *ARF17,* is required to develop many organs, particularly in embryos [283]. Therefore, *MIR160a* loss-of-function mutants exhibit various defects during embryogenesis [284]. By contrast, transgenic expression of a *miR160*-resistant form of *ARF10* was associated with a high level of shoot regeneration. This transgenic line also showed an elevated expression level of shoot meristem-specific genes *CL3*, *CUC1* and -*2*, and *WUS* [280].

In *Arabidopsis*, *miR165/166* and *miR160* contribute to the induction of somatic embryogenesis associated with the *LEC2*-controlled auxin response that occurs in somatic cells during the embryogenic transition [275]. *MiR167* functions in somatic embryogenesis by regulating the expression of its target auxin response genes [285]. Thus, specific miRNAs target the genes that govern the transition from differentiated to totipotent cells. The differential expression of miRNAs during somatic embryogenesis has also been observed in other plant species, such as *Oryza sativa* [286] and *Zea mays* [287].

ARGONAUTE (AGO) proteins bind with mature miRNAs to guide the riboprotein complex to its target mRNA [288,289]. In vitro, AGO10 inhibited shoot regeneration via repressing de novo SAM formation [286]. In fact, in in vitro cultured explants of the loss-of-function mutant *ago10*, a much larger number of SAMs was formed, and within these, the stem cell marker genes *WUS*, *CLV3*, and *STM* were all strongly expressed [290]. *AGO10* repressed the accumulation of the *miR165*/*166*, thereby upregulating *HD-ZIP III* genes. The overproduction of *miR166* was shown to promote shoot regeneration. At the same time, the absence of *miR165*/*166* information resulted in a blockage to shoot regeneration, and only a partial rescue of the phenotype of the *ago10* mutant [290]. Notably, explants derived from the *men1* mutant (an overproducer of *miR166a*) regenerated shoots more readily than WT explants, while explants derived from the loss-of-function of the *miR165*/*166* line *MIM165*/*166* were less productive [290]; these observations suggest that *miR165*/*166* acted to promote adventitious shoots.

Thus, epigenetic mechanisms appear to be essential upstream factors in determining embryogenic capacity; during the induction of somatic embryogenesis, the remodeling of chromatin results in the release of the embryogenic program otherwise repressed by chromatin-based silencing mechanisms. Consequently, variations in the expression profile of genes directly involved in the epigenetic regulation of cell transcriptome could have a primary role in initiating an embryogenic program.

## 7. Conclusions and Prospects

In conclusion, as underlined by Jha et al. [88], *WOX* genes are master regulators of different aspects of plant biology and in Figure 4 a simplified overview is reported. We focused on the involvement of these genes in the in vitro regeneration of plants and, more generally, in the manifestation of cellular totipotency.

In particular, somatic embryogenesis is a fascinating phenomenon in plant biology, and is useful for vegetative propagation and plant regeneration in genome editing methods.

From the pioneering studies conducted with in vitro experiments, information about the nature of the factors underlying the induction processes has dramatically increased. The pathway to developing SE from somatic cells is not always the same, and, as suggested by Fehér [35], the claim to identify a key trigger valid for all somatic embryogenic systems will be challenging to achieve. In several plants, differential expression of specific genes is observed in vitro during the induction of SE; however, to date, the hierarchical relationship between the genes found to be differentially expressed is only partially decoded, depending on the analyzed species. In *Arabidopsis*, an updated hierarchical mechanism for SE in vitro has recently been suggested [45]. Regarding this, at the top is a permissive chromatin environment to allow the reactivation of genes encoding cell totipotency-related TFs with a concurrent auxin stimulation. Hence, at the successive step, direct activation of early embryonic development genes such as *WOX2* and *WOX3* occurs [45]. Concerning the modification of the epigenetic landscape, one of the more critical genes is *PRC2* [245,291]. At the same time, regarding the totipotency-related TFs, a position of absolute prominence should undoubtedly be assigned to BBM [43].

What about WUS in the induction of SE? The somatic embryogenesis is unaffected in the *wus* mutant of *Arabidopsis*, although malformations in the shoot meristem were recognized [80]. Furthermore, numerous studies have shown that *WUS* expression is often induced during the induction of SEs, and overexpression is associated with a more significant SE differentiation in some species [88,292]. In addition, the concomitant overexpression of *BBM* and *WUS* in monocots promotes SE induction, with a consequent increase in *Agrobacterium*-mediated transformation percentage [174]. However, the involvement of the *WUS* gene in regeneration processes in vitro cannot be confined to the embryogenic pathway, since its involvement in the differentiation of adventitious buds was repeatedly observed. Indeed, during the induction of shoots in vitro from hypocotyl explants in *Arabidopsis*, transcripts of the *WUS* gene mark the shoot progenitor region of cells at an extremely early stage [62]. It is interesting to point out that for the ectopic *WUS* expression in a restricted group of cells, a sequence of factors is required: exogenous cytokinin supply to remove a repressive condition of epigenetic nature, and then the action of ARRs genes and binding with microRNA165/6-targeted HD-ZIP III TFs [62]. Cytokinin signaling and a permissive epigenetic environment are also required in de novo activation of *WUS* expression during axillary meristem initiation in vivo [293]. To date, no such detailed chain of molecular events has described the role of the WUS gene in the induction of somatic embryogenesis. Zuo et al. [70] have obtained strong evidence regarding the importance of *WUS* in the vegetative-to-embryonic transition in *Arabidopsis*. However, the authors have not characterized the *WUS*-expressing cells at the morphological and functional levels, and it is still unclear whether that group of cells represents a functional organizing center [70]. On the other hand, WUS represses *LEC1* expression, suggesting that WUS cannot activate the embryo identity pathway [70].

Furthermore, it should be noted that in vivo, *WUS* and *WOX* genes are also implicated in vegetative propagation, such as epiphyllous plantlets and the differentiation of bulbils [135,136]. In particular, somatic embryogenesis in vivo is a remarkable phenomenon, but, unfortunately, it has not been studied in detail. It might be interesting to learn, in more depth, the role of *WOX* genes in epiphylly, since only a few genes have been investigated at present [26,294,295,296,297]. In this regard, the recent study by Jácome-Blásquez et al. [298] appeared to increase knowledge regarding the genes involved in the ectopic differentiation of plantlets in vivo. In addition, it might also be of interest to address the molecular characterization of somatic embryogenesis in other plant species with a spontaneous ability to develop in vivo ectopic embryos, such as *Malaxis paludosa* [22]. We are still far from outlining a model that links, in a hierarchical relationship, the crucial factors for natural SE.

## Figures and Tables

**Figure 1 ijms-23-15950-f001:**
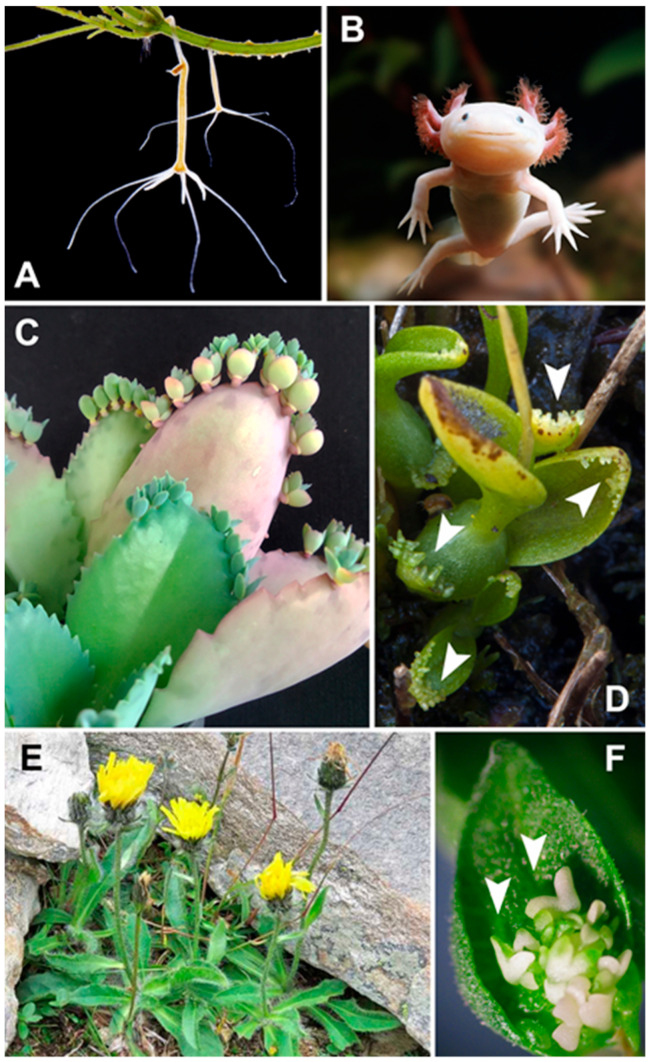
Animals able to regenerate lost body organs in vivo: (**A**) *Hydra vulgare*, an aquatic invertebrate (©Warren Photographic, www.warrenphotographic.co.uk/05836-common-hydra-budding; accessed on 11 October 2022). (**B**) Mexican axolotl (*Ambystoma mexicanum*), an aquatic salamander (©Warren Photographic, www.warrenphotographic.co.uk/00108-albino-axolotl; accessed on 11 October 2022). Examples of plants with natural epiphylly: (**C**) *Kalanchoe laetivirens*. (**D**) *Malaxis paludosa* (Michael Dodd’s photo, Open University, Milton Keynes, UK). The white arrowheads indicate the propagules. (**E**) *Hieracium alpinum*, a species of *Asteraceae* with an apomictic triploid cytotype in vivo (Mauro Felicioli’s photo, www.actaplantarum.org/flora/flora_info.php?id=506593; accessed on 11 October 2022). (**F**) An unusual example of a plant with epiphyllous embryos induced in vitro: the interspecific hybrid EMB-2 (*Helianthus annuus* x *H. tuberosus*). The white arrowheads indicate the somatic embryos.

**Figure 2 ijms-23-15950-f002:**
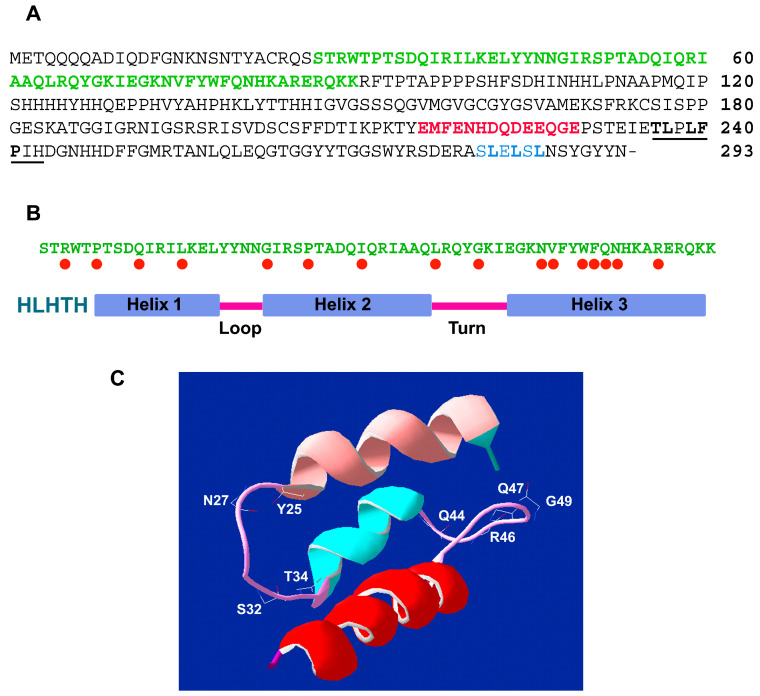
Analysis of WUSCHEL transcription factor from sunflower (HaWUS) (GenBank accession HE616565.1). (**A**). Amino acid sequence of HaWUS (293 amino acid residues). The homeodomain is in bold and green characters; the WUS-box motif (**TL**P**LFP**XX) is underlined, and the conserved amino acid residues are in bold character. The EAR-like motif is in light blue character, the conserved amino acid residues are in bold character, and the acidic region is in red character [84]. (**B**). Within the homeodomain, the helix-loop-helix-turn-helix (HLHTH) motifs are indicated. Red dots evidence completely conserved residues. (**C**). Modeling of the three-dimensional structure of HaWUS in the homeodomain. The predicted modeling was generated by GENO3D (http://geno3d-pbil.ibcp.fr; accessed on 10 January 2015) and displayed by Deep View/Swiss Pdb-viewer v.4.1 (http://www.expasy.org/spdbv/; accessed on 10 January 2015).

**Figure 3 ijms-23-15950-f003:**
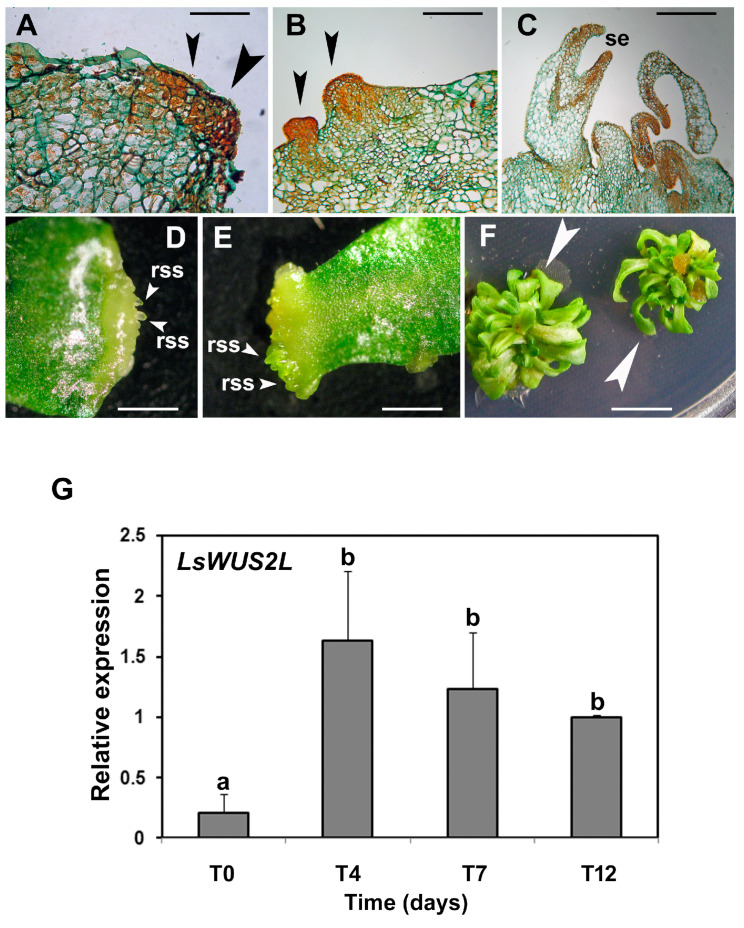
In vitro regeneration of *Lactuca sativa* cv. ‘Romana’ on a regeneration medium with 2.2 μM of 6-benzylaminopurine and 0.54 μM of α-naphthalenacetic acid; time-dependent analysis of *WUSCHEL2-LIKE* (*LsWUS2L*) expression levels. (**A**–**C**): Histological analysis of cotyledon explants cultivated in vitro. Explants sectioned after 4 [T4 (**A**)], 7 [T7 (**B**)], and 12 [T12 (**C**)] days of culture on the regeneration medium. (**A**) Longitudinal section of a callus area characterized by small and poorly vacuolated cells located in a marginal region (black arrowheads). (**B**). Longitudinal section of an explant with initial stages of adventitious structures (black arrowheads). (**C**). Longitudinal section of an explant with early-stage somatic embryos (se). Scale bars: 0.7 mm (**A**), 0.19 mm (**B**), 0.28 mm (**C**). Binocular microscope photos of cotyledon explants in regeneration. (**D**), (**E**) Explants after 7 days (T7) of culture on regeneration medium: regeneration events occurred with the appearance of round-shaped structures (rss, white arrowheads) characterized by a regular profile. (**F**) Explants with massive formation of well-organized regenerated plantlets (white arrowheads). Scale bars: 0.85 mm (**D**), 1.02 mm (**E**), 8.7 mm (**F**). (**G**) Time-course analysis of the expression of *WUSCHEL2-LIKE* (*LsWUS2L*) during in vitro regeneration from cotyledon explants of lettuce. The times of in vitro culture analyzed were: T0 (0 days), T4 (4 days), T7 (7 days), and T12 (12 days). The relative transcriptional values were calculated using T12 as the reference sample. Data are means ± standard deviation (SD) (*n* = 4). Different letters indicate statistically significant differences between genotypes (one-way ANOVA and Tukey’s test: *p* < 0.05) [69].

**Figure 4 ijms-23-15950-f004:**
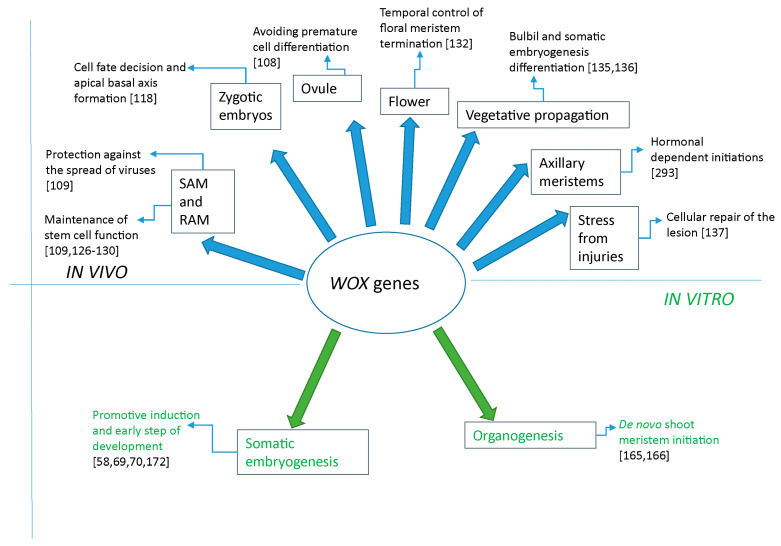
Schematic representation of some major roles of *WOX* genes in vivo and in vitro in plants. The diagram is simplified, and bibliographic references are partial.

**Table 1 ijms-23-15950-t001:** Involvement of *WUS* and *WOX* genes in somatic embryogenesis and organogenesis in vitro, observed in different herbaceous and tree plants.

Gene	Species	Molecular Event Analyzed	Process Involved	Reference
*AtWUS*	*Arabidopsis thaliana*	Overexpression	PGR independent somatic embryo development	[70]
*AtWUS*	*Arabidopsis thaliana*	Overexpression	Regeneration in PGR-free medium	[62]
*AtWUS* and *AtWOX5*	*Arabidopsis thaliana*	Start of expression	In cells of callus, previous events of ectopic morphogenesis	[158]
*PgWOX2*	*Picea glauca*	Start of expression	During the early stage of somatic embryogenesis	[168]
*LdWOX2*	*Larix decidua*	Start of expression	During the early stage of somatic embryogenesis	[169]
*PpWOX2*	*Pinus pinaster*	High expression	During somatic embryogenesis proliferation	[170]
*PpWOX2*	*Pinus pinaster*	Overexpression	Negative effect on the maturation of somatic embryos	[170]
*VvWOX2* and *VvWOX9*	*Vitis vinifera*	Start of expression	During the early stage of somatic embryogenesis	[58]
*VvWOX3* and *VvWOX11*	*Vitis vinifera*	High expression levels	Specific stages of somatic embryos (torpedo and cotyledonary)	[58]
*MtWOX11-like*	*Medicago truncatula*	Start of expression	During the early stage of somatic embryogenesis	[159]
*MtWUS* and *MtWOX5*	*Medigago truncatula*	Start of expression	In calli with different embryogenic competence	[171]
*MtWUS*	*Medicago truncatula*	Expression cytokinin-dependent	During the induction of totipotent stem cells	[172]
*CjWUS*	*Cryptomeria japonica*	Upregulation of expression	In callus with a high rate of embryogenesis	[173]

## Data Availability

Not applicable.

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
