# Peer review of "Induction of Somatic Embryogenesis in Plants: Different Players and Focus on WUSCHEL and WUS-RELATED HOMEOBOX (WOX) Transcription Factors"

_ijms, 2022, doi:10.3390/ijms232415950_

Round 1
Reviewer 1 Report
Overall, I found this to be a very nice review of induction of somatic embryogenesis (SE) and the role that the WOX transcription factors (TFs) may play in this process. The literature reviewed is current with more than one-half in the past near decade (2010 plus).
There are a number of issues with English –some examples but not comprehensive:
Lin 32: “concerning” or “compared to”
Lines 87-90ish – italicize gene names.
Line 110: needs (not need)
Line 136: helix (not elix).
And so on, minor cleanup is needed.
My next comment – this is a very nice and current review with (by my quick count) more than one-half of the citations since 2010. However, this is very text heavy. Are there ways to add models (for example, pg 7-8 on the interaction of cytokinin and WINDs… sure I realize this would be a working model, but I think still helpful? Another example comes up on p. 9 – could you make a table to summarize WOX effects across different species? Yet another visual that could help – p. 10: could you generate and show the heat map for lines 380-388?
I think you can see my main comment – some visuals would help the reader.
I was also not totally sure on the link between epigenetics and WUS, including BBM. I think I missed something here, but again.. super text heavy. Maybe a working model?
Just some thoughts to try to make a nice comprehensive review even easier (for this reviewer) to appreciate.
Author Response
Reviewer 1
Comments and Suggestions for Authors
Overall, I found this to be a very nice review of induction of somatic embryogenesis (SE) and the role that the WOX transcription factors (TFs) may play in this process. The literature reviewed is current with more than one-half in the past near decade (2010 plus).
There are a number of issues with English –some examples but not comprehensive:
Lin 32: “concerning” or “compared to”
Our Reply: we apologize for the mistake; “compared to” has been included. More generally, as the Reviewer suggested, we thoroughly check the English.
Lines 87-90ish – italicize gene names.
Our Reply: With respect “WUSCHEL (WUS), AINTEGUMENTA-LIKE (AIL) AP2/ERF-domain, etc.” we intended to indicate the proteins encoded by the genes, and thus the Transcription Factors were not written in italics.
Line 110: needs (not need)
Our Reply: we apologize for the mistake; “needs” has been included.
Line 136: helix (not elix).
Our Reply: we apologize for the mistake; “helix” has been included.
And so on, minor cleanup is needed.
Our Reply: as required, a careful analysis of the text was carried out.
My next comment – this is a very nice and current review with (by my quick count) more than one-half of the citations since 2010. However, this is very text heavy. Are there ways to add models (for example, pg 7-8 on the interaction of cytokinin and WINDs… sure I realize this would be a working model, but I think still helpful? Another example comes up on p. 9 – could you make a table to summarize WOX effects across different species? Yet another visual that could help – p. 10: could you generate and show the heat map for lines 380-388?
Our Reply: About the 'heaviness of the text', we have eliminated about fifty lines of text in the revised version; the latter has been lightened. In addition, several paragraphs have been moved to the new sub-paragraphs inserted.
Concerning the phrases relative to WOX13 and WOUND-INDUCED DEDIFFERENTIATION 1 (WIND1) interaction, these have been moved from their original position and inserted in a new sub-paragraph “2.2 In Vivo Roles of WOX Genes in Plants: Developmental Aspects and Stress Response”. We consider this new location more appropriate, which also improves understanding. Furthermore, we added a new phrase to recall the reference to Ikeuchi et al.’s schematic diagram of WOX13 function in callus formation and tissue repair. Finally, unfortunately, we do not precisely understand the meaning of Reviewer 1's note on the “interaction between WIND and cytokinins”. The interaction between WIND and cytokinins was not investigated in the citation [137] (Ikeuchi et al. 2022). On the other hand, Iwase et al. (2011; Plant Signal. Behav. 6(12):1943-5. doi: 10.4161/psb.6.12.18266) showed that WIND proteins are required to activate the local cytokinin response at the wound site. However, this report was not included in our text.
Following the Reviewer’s opinion, a new Table 1 has been included in the revised manuscript to summarize the WOX roles in morphogenesis. We hope this choice will help lighten the text's heaviness.
Regarding Reviewer's Note 1 concerning lines 380 to 388, we have included the results of the relative citation [173] in Table 1.
I think you can see my main comment – some visuals would help the reader.
Our Reply: Accordingly, a new Table 1 and Figure 4 have been included in the revised version.
I was also not totally sure on the link between epigenetics and WUS, including BBM. I think I missed something here, but again.. super text heavy. Maybe a working model?
Our Reply: In the revised version of our manuscript, the old Paragraph 4, “An Epigenetic Hierarchical Network Reprogam Differentiates Cells to Pluripotent/Totipotent Cells”, has been renovated. A new Paragraph, “Molecular players in plant totipotency”, has been included, and in this, some concepts regarding WUS and BBM are present.
Also, in the revised version of the manuscript, a modified form of the Paragraph “An Epigenetic Hierarchical Network Reprogam Differentiates Cells to Pluripotent/Totipotent Cells” (now, number 6) is present because the resetting of the epigenetic landscape during cell fate reprogramming has a basic role.
Just some thoughts to try to make a nice comprehensive review even easier (for this reviewer) to appreciate.
Our Reply: We agree with the reviewer and hope to have come close to that goal in the revised version of the manuscript.
Reviewer 2 Report
This manuscript major reviews that the relationship between induction of somatic embryogenesis and WUSCHEL gene. This topic is very interesting and useful. However, the overall structure of this article is a little messy, and the logic needs to be strengthened. The detail suggestions are shown:
1、Line 45: The word "several" is a bit vague, please provide some detail gene names.
2、In the second section of “WUS-Related Homeobox (WOX) Genes”, this part of the author's writing is too messy. In this part, either only the gene classification of WUS is written, or a brief introduction is made according to the sequence of gene names.
3、Line 270-276: This part has no relationship with the theme of this article and should be deleted. The review is not a simple list of references, but a logical introduction according to the theme. There are also many sentencse are no useful for this manuscript, please carefully select and organizations.
4、Lines 334-395: this section under the title of “Interaction of WOX Genes with Hormones and the Establishment of Embryogenic Totipotency”, but there are no relationships between them. The author can use more subtitles to introduce.
5、The author introduces where many genes are expressed (Lines 330-356; lines 556-570, etc. One or two references like this can be enough. The focus is on the function of genes. For example, the overexpression TaWOX5 can increase the transformation efficiency of wheat (Wang et al. 2022).
6、Lines 475-495, this section is also very few related with this theme.
7、It would be better that drawing a map of the relationship between Wus gene and SE according to the published literature.
Reference:
Wang K, Shi L, Liang XN, Zhao P, Wang WX, Liu JX, Chang YN, Hiei Y, Yanagihara C, Du LP, Ishida Y, Ye XG. The gene TaWOX5 overcomes genotype dependency in wheat genetic transformation. Nature Plants, 2022,8:110-117.
Author Response
Reviewer 2
Comments and Suggestions for Authors
This manuscript major reviews that the relationship between induction of somatic embryogenesis and WUSCHEL gene. This topic is very interesting and useful. However, the overall structure of this article is a little messy, and the logic needs to be strengthened. The detail suggestions are shown:
1) Line 45: The word "several" is a bit vague, please provide some detail gene names.
Our Reply: In agreement with the Reviewer, details have been included in the text.
2) In the second section of “WUS-Related Homeobox (WOX) Genes”, this part of the author's writing is too messy. In this part, either only the gene classification of WUS is written, or a brief introduction is made according to the sequence of gene names.
Our Reply: Accordingly with the Reviewer, we divided the old Paragraph “WUS-Related Homeobox (WOX) Genes” into two new Sub-Paragraphs (2.1 Nomenclature and Molecular Motifs Characteristics; 2.2 In Vivo Roles of WOX Genes in Plants: Developmental Aspects and Stress Response) to better order the topics we intended to develop.
Although the title of our manuscript focuses on the induction of somatic embryogenesis, it is interesting to briefly summarize some of the roles in vivo that WOX genes play for the plant.
3) Line 270-276: This part has no relationship with the theme of this article and should be deleted. The review is not a simple list of references, but a logical introduction according to the theme. There are also many sentencse are no useful for this manuscript, please carefully select and organizations.
Our Reply: We have contextualised this part in the new sub-section 2.2, in which some of the roles exerted by WOX genes in plants in vivo are discussed.
Specifically, repair processes in plant tissue are, in our opinion, a remarkable phenomenon also with applicative interest. Therefore, the results recently presented by Ikeuchi et al. (2022) [137] are in line with the new sub-paragraph because the involvement of WUSCHEL-RELATED HOMEOBOX 13 was demonstrated. In addition, the WIND1 gene also plays a role in cell fate switch in vitro.
4) Lines 334-395: this section under the title of “Interaction of WOX Genes with Hormones and the Establishment of Embryogenic Totipotency”, but there are no relationships between them. The author can use more subtitles to introduce.
Our Reply: As noted above, this part of the manuscript has been profoundly modified by including Table 1. The criterion that guided us in discussing some research in more detail in the text rather than others was to select the in vitro morphogenesis research in which the hormone-WOX interaction was really addressed.